EMBO
Molecular Medicine

# Cerebrospinal fluid and plasma biomarker trajectories with increasing amyloid deposition in Alzheimer's disease

Sebastian Palmqvist[1,2,*] (iD), Philip S Insel[1,3], Erik Stomrud[1,4], Shorena Janelidze[1], Henrik Zetterberg[5,6,7,8], Britta Brix[9], Udo Eichenlaub[10], Jeffrey L Dage[11], Xiyun Chai[11], Kaj Blennow[5,6], Niklas Mattsson[1,2,12] (iD) & Oskar Hansson[1,4,**] (iD)

## Abstract

Failures in Alzheimer's disease (AD) drug trials highlight the need to further explore disease mechanisms and alterations of biomarkers during the development of AD. Using cross-sectional data from 377 participants in the BioFINDER study, we examined seven cerebrospinal fluid (CSF) and six plasma biomarkers in relation to β-amyloid (Aβ) PET uptake to understand their evolution during AD. In CSF, Aβ42 changed first, closely followed by Aβ42/Aβ40, phosphorylated-tau (P-tau), and total-tau (T-tau). CSF neurogranin, YKL-40, and neurofilament light increased after the point of Aβ PET positivity. The findings were replicated using Aβ42, Aβ40, P-tau, and T-tau assays from five different manufacturers. Changes were seen approximately simultaneously for CSF and plasma biomarkers. Overall, plasma biomarkers had smaller dynamic ranges, except for CSF and plasma P-tau which were similar. In conclusion, using state-of-the-art biomarkers, we identified the first changes in Aβ, closely followed by soluble tau. Only after Aβ PET became abnormal, biomarkers of neuroinflammation, synaptic dysfunction, and neurodegeneration were altered. These findings lend *in vivo* support of the amyloid cascade hypotheses in humans.

**Keywords** Alzheimer disease; amyloid positron emission tomography; cerebrospinal fluid biomarkers; plasma biomarkers
**Subject Categories** Biomarkers; Neuroscience

## Introduction

Continued failures in clinical trials for Alzheimer's disease (AD) against presumably the correct drug targets highlight the need for further research to understand all important mechanisms and at what stage they occur and become measurable (Honig *et al*, 2018; Jack *et al*, 2018; Egan *et al*, 2019; Knopman, 2019; Selkoe, 2019). The pathogenesis of AD is complex and involves many different mechanisms. According to the amyloid cascade hypothesis, the first hallmark pathology of AD is the abnormal accumulation of β-amyloid (Aβ) that can start decades before the dementia stage and continues throughout the course of the disease (Villemagne *et al*, 2013). Aβ is thought to trigger or drive tau pathology, which, possibly together with inflammatory mechanisms, may cause synaptic dysfunction and neurodegeneration that result in cognitive impairment and dementia (Jack *et al*, 2013; Sperling *et al*, 2014).

There are now several cerebrospinal fluid (CSF) and plasma biomarkers that, to different extent, measure these different pathogenic mechanisms. Examining these, especially in the earlier stages of AD, would allow us to better understand the pathogenesis of the disease, which is essential for identifying potential drug targets, designing clinical trials, and improving diagnostics and the clinical work-up of AD (Blennow *et al*, 2010). By

1  Clinical Memory Research Unit, Department of Clinical Sciences, Lund University, Lund, Sweden
2  Department of Neurology, Skåne University Hospital, Lund, Sweden
3  Department of Psychiatry, University of California, San Francisco, San Francisco, CA, USA
4  Memory Clinic, Skåne University Hospital, Malmö, Sweden
5  Department of Psychiatry and Neurochemistry, The Sahlgrenska Academy at the University of Gothenburg, Mölndal, Sweden
6  Clinical Neurochemistry Laboratory, Sahlgrenska University Hospital, Mölndal, Sweden
7  Department of Neurodegenerative Disease, UCL Institute of Neurology, Queen Square, London, UK
8  UK Dementia Research Institute at UCL, London, UK
9  Euroimmun AG, Lübeck, Germany
10 Roche Diagnostics GmbH, Penzberg, Germany
11 Eli Lilly and Company, Indianapolis, IN, USA
12 Wallenberg Center for Molecular Medicine, Lund University, Lund, Sweden
   *Corresponding author. Tel: +46 46 177808; E-mail: sebastian.palmqvist@med.lu.se
   **Corresponding author. Tel: +46 40-335036; Fax: +46 40-335657; E-mail: oskar.hansson@med.lu.se
   ELECSYS, COBAS, and COBAS E are registered trademarks of Roche. All trademarks mentioned enjoy legal protection

comparing CSF and plasma biomarkers, we can also understand which disease mechanisms we can identify by measuring biomarkers in blood instead of in CSF samples.

In this study of 377 elderly, non-demented participants from the BioFINDER study, we examined CSF and plasma biomarkers for Aβ (Aβ42 and Aβ40), tau (P-tau), synaptic dysfunction (neurogranin), neurodegeneration [total-tau (T-tau) and neurofilament light chain (NfL)], and glial activation and neuroinflammation (YKL-40, measured only in CSF). The biomarker changes were modeled as functions of the Aβ positron emission tomography (PET) signal that measures the amount of accumulated fibrillar Aβ in the neocortex (used as a proxy for time in the disease). We identified at what Aβ load significant changes in the biomarkers occurred (change point of the trajectory). Differences between the change points of the biomarkers were examined to map the temporal evolution of the biomarkers. Finally, CSF Aβ42, Aβ40, T-tau, and P-tau assays from five different manufacturers were compared to assess the generalizability of the results.

# Results

Demographic and clinical data for the study participants are shown in Table 1. Of the 377 included participants, 242 were cognitively unimpaired (CU) and 135 had mild cognitive impairment (MCI). According to the mixture modeling-derived Aβ PET cutoff of < 0.736 SUVR, 151 participants (40%) were Aβ-positive (Aβ+) and 226 (60%) Aβ-negative (Aβ−). All plasma and CSF biomarkers were significantly different between Aβ+ and Aβ− participants, except for CSF Aβ40, plasma Aβ40, plasma tau, and plasma neurogranin.

## CSF biomarker trajectories as a function of increasing Aβ accumulation

The monotone spline models of the CSF biomarkers are shown in Fig 1A. Separate biomarker models with data points, significance level, and $r^2$ value are shown in Appendix Fig S1. Note that all models were fitted using cross-sectional CSF, plasma, and PET data.

**Table 1. Demographic and clinical data stratified by Aβ positivity.**

| Variable | Total population | Aβ+ | Aβ− | P-value |
|---|---|---|---|---|
| N | 377 | 151 | 226 | |
| Age (years) | 72.1 (5.4) | 72.6 (5.0) | 71.8 (5.6) | 0.10 |
| Sex (female) | 50% | 44% | 54% | **0.042** |
| MMSE (0–30 points) | 28.3 (1.6) | 27.8 (1.6) | 28.5 (1.5) | **<0.001** |
| APOE ε4-positive | 38% | 66% | 22% | **<0.001** |
| Aβ PET (SUVR)[a] | 0.782 (0.23) | 1.023 (0.18) | 0.622 (0.05) | **<0.001** |
| Hippocampus volume/ICV | 0.0045 (0.00069) | 0.00425 (0.00062) | 0.00468 (0.00068) | **<0.001** |
| CSF biomarker (pg/ml) | | | | |
| Aβ42 | 1,321 (650) | 818 (319) | 1,657 (596) | **<0.001** |
| Aβ40 | 22,811 (82,293) | 29,261 (129,856) | 18,501 (5,362) | 0.57 |
| Aβ42/Aβ40 | 0.0717 (0.028) | 0.0448 (0.0164) | 0.0898 (0.0187) | **<0.001** |
| T-tau | 256 (116) | 319 (139) | 215 (73.0) | **<0.001** |
| P-tau | 22.8 (12.4) | 30.0 (15.1) | 17.9 (6.7) | **<0.001** |
| NfL | 1,192 (948) | 1,399 (1,133) | 1,053 (847) | **<0.001** |
| Neurogranin | 405 (213) | 480 (253) | 356 (164) | **<0.001** |
| YKL-40 | 194,090 (63,108) | 205,273 (64,958) | 186,618 (60,847) | **0.003** |
| Plasma biomarker (pg/ml) | | | | |
| Aβ42 | 31.6 (4.9) | 29.9 (4.7) | 32.7 (4.7) | **<0.001** |
| Aβ40 | 484 (72) | 483 (73) | 485 (71) | 0.66 |
| Aβ42/Aβ40 | 0.0657 (0.0082) | 0.0622 (0.0078) | 0.0680 (0.0077) | **<0.001** |
| T-tau | 17.8 (5.3) | 18.2 (5.0) | 17.6 (5.5) | 0.12 |
| P-tau | 2.7 (4.6) | 3.4 (3.2) | 2.1 (5.3) | **<0.001** |
| NfL | 22.9 (17.0) | 23.9 (11.2) | 22.2 (19.9) | **0.003** |
| Neurogranin | 20,205 (10,655) | 19,414 (10,961) | 20,735 (10,437) | 0.17 |

Values are in mean (SD) if not otherwise stated. Mann–Whitney was used to compare the Aβ+ and Aβ− groups.
Bold P-values indicate statistical significance at P < 0.05.
CU, cognitively unimpaired; ICV, intra cranial volume; MCI, mild cognitive impairment; MMSE, Mini-Mental State Examination; N, number of participants; NfL, neurofilament light chain; P-tau, phosphorylated-tau; SD, standard deviation, T-tau; total-tau.
[a]Early Aβ accumulating ROI with a composite reference region (see Materials and Methods).

**A**   CSF biomarker models

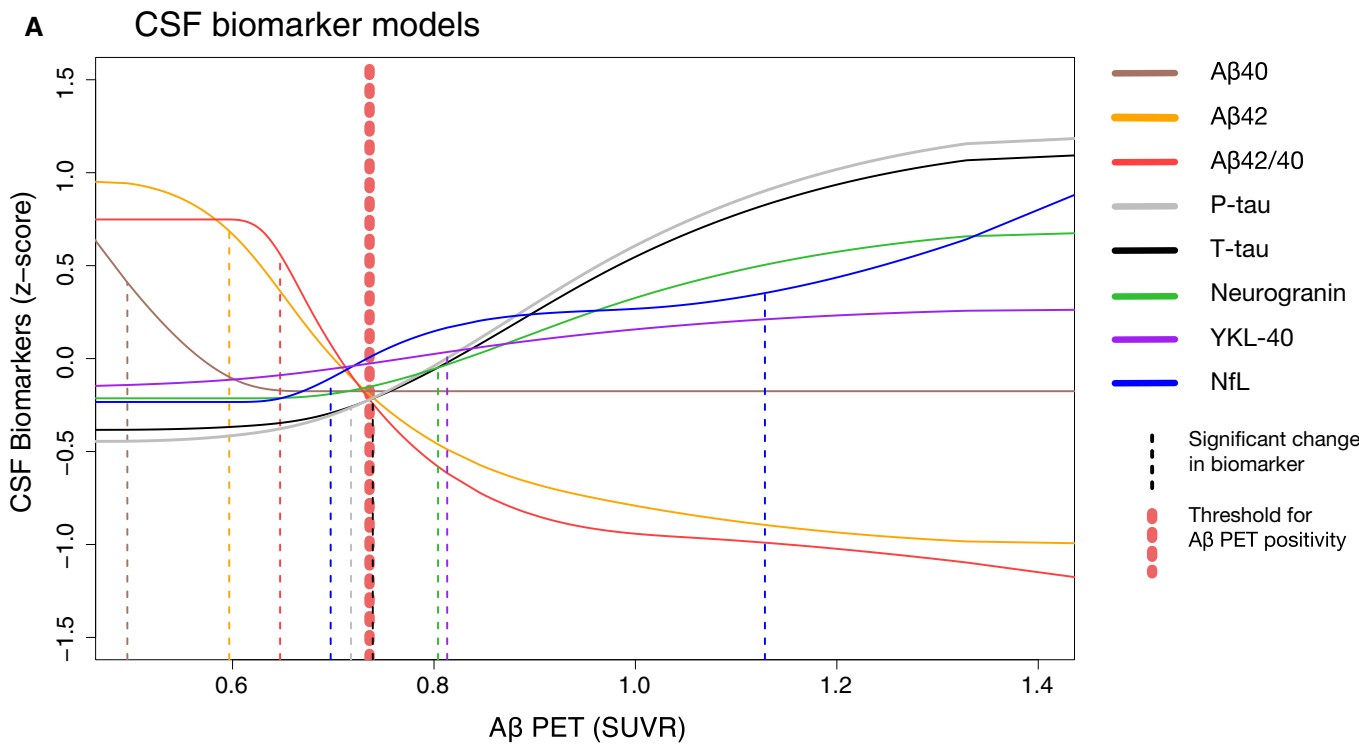

**B**   Plasma biomarker models

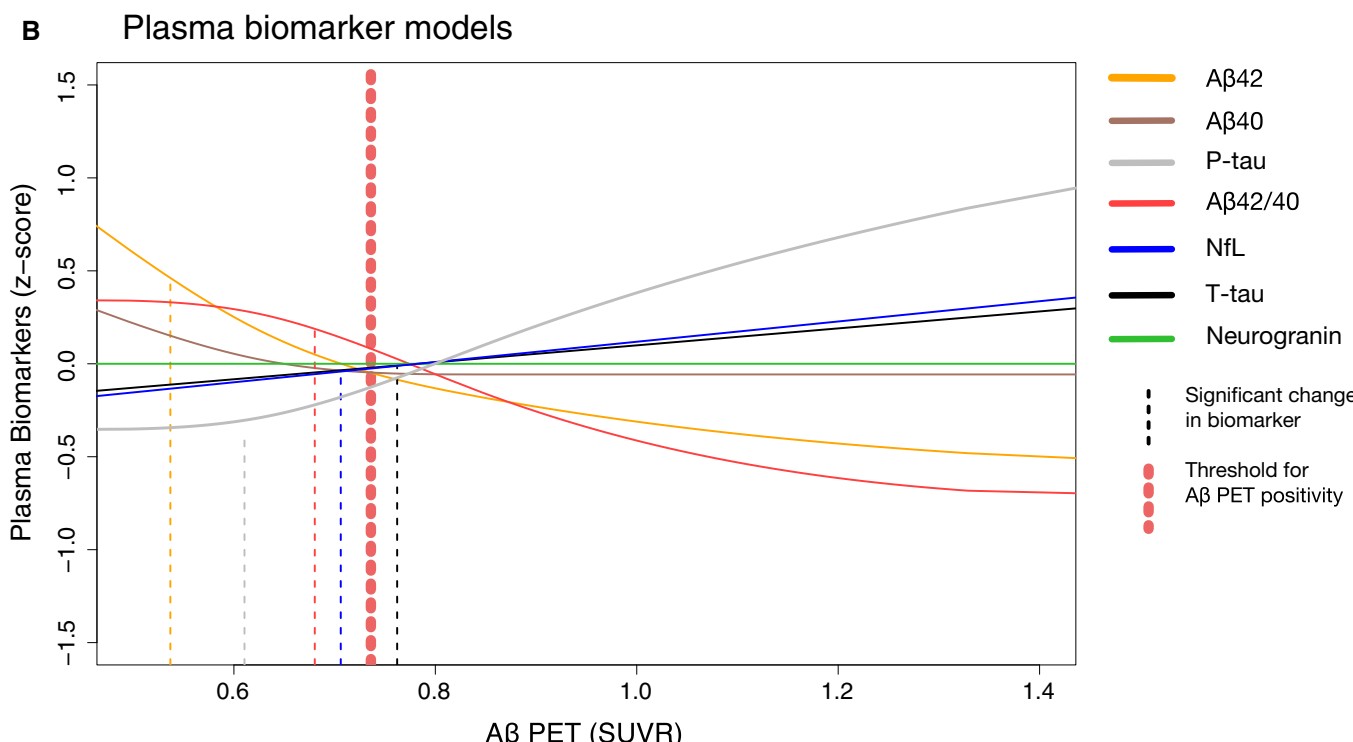

**Figure 1.   CSF and plasma biomarker trajectories as a function of increasing Aβ accumulation.**

A, B   The biomarker data were fitted using monotone spline models where Aβ PET SUVR acted as a proxy for time in AD. CSF (A) and plasma (B) biomarkers are shown separately, but selected direct comparisons are shown in Fig 3. Individual spline models with actual data points are shown in Appendix Figs S1 and S2. The threshold for Aβ was established using mixture modeling statistics. Point of change on the trajectory (also referred to as significant biomarker change) is shown as vertical dashed lines. They were defined as a change in 2 SE (derived from 500 bootstrap samples) from the starting point of the modeled trajectory. Note that plasma P-tau data were missing in 34 cases. To facilitate comparisons between different CSF and plasma biomarkers, the levels have been transformed to z-scores based on the distribution in the present population (i.e., a z-score of 0 corresponds to the mean of the study cohort).

The models were significantly fitted for all the CSF biomarkers. Initial declines were seen for both CSF Aβ42 and Aβ40, followed by a flat curve for Aβ40 (i.e., no association with further increases in SUVR) while Aβ42 continued to decrease after the level when Aβ positivity was reached. In concordance with this, the CSF Aβ42/Aβ40 ratio started with a plateau, followed by a later drop compared to Aβ42 (note that no increase in Aβ42/Aβ40 would be estimated given the *a priori* assumption of monotonicity of Aβ42/Aβ40 with respect to SUVR). Around 1.0 SUVR (after Aβ positivity was reached), both Aβ42 and Aβ42/40 flattened out and did not continue to decline as Aβ PET SUVR increased further. CSF T-tau and P-tau had very similar trajectories with the greatest increase after Aβ positivity was reached. CSF neurogranin showed a more modest increase and smaller dynamic range throughout the SUVR span, and the change was even more modest for CSF YKL-40 (< 0.5 $z$-score). CSF NfL exhibited a sigmoid trajectory with a subtle initial increase around the same point as most other CSF biomarkers, followed by a plateau, and then a marked increase at a later stage that continued to increase throughout the span of SUVRs (Fig 1A). Based on this appearance, two change points were established for CSF NfL (see below and Materials and Methods). As expected when using Aβ PET as the dependent variable, the best model fits were seen for CSF Aβ42 ($r^2 = 0.42$) and CSF Aβ42/Aβ40 ($r^2 = 0.55$), while poorer fits were seen for CSF P-tau ($r^2 = 0.30$), T-tau ($r^2 = 0.25$), neurogranin ($r^2 = 0.11$), NfL ($r^2 = 0.11$), YKL-40 ($r^2 = 0.02$), and Aβ40 ($r^2 = 0.02$).

### CSF biomarker change points

A significant biomarker change (or "change point") was defined as a 2 standard error (SE) change from the starting point of the spline based on 500 bootstrap samples. These change points are marked as vertical dashed lines in Fig 1 and shown with 95% CIs in Fig 2 and in Appendix Table S1. The first significant changes for CSF biomarkers were seen for Aβ40, followed by Aβ42 and then Aβ42/Aβ40. Later, increases in P-tau, T-tau, and NfL were seen, with no significant differences between them (i.e., overlapping 95% CIs). These latter biomarker changes occurred just before or at the time of Aβ positivity. Slightly later, changes in neurogranin, YKL-40, and hippocampal volume were seen. The second increase in NfL occurred last and significantly later than all other biomarker changes.

### Plasma biomarker trajectories

Plasma biomarker splines are shown in Fig 1B. As for the plasma biomarker models, all were significant except for neurogranin and Aβ40 (Appendix Fig S2). Similar to CSF Aβ42 and Aβ40, plasma Aβ42 and Aβ40 showed an initial, parallel decline followed by a flat line for Aβ40 resulting in a later drop for the Aβ42/40 ratio (Fig 3A). In contrast to CSF, plasma Aβ42 and Aβ42/40 showed more modest changes over the entire Aβ accumulation range (about 1 $z$-score vs. about 2 $z$-scores for CSF; Fig 3A) and had overall a lesser agreement with Aβ PET (plasma $r^2$: 0.07–0.12; CSF $r^2$: 0.42–0.55; Appendix Figs S1 and S2). This lesser agreement was true for all plasma biomarkers compared with the corresponding CSF biomarkers (Appendix Figs S1 and S2), except for plasma P-tau which was more similar to the corresponding CSF biomarker (Fig 3B).

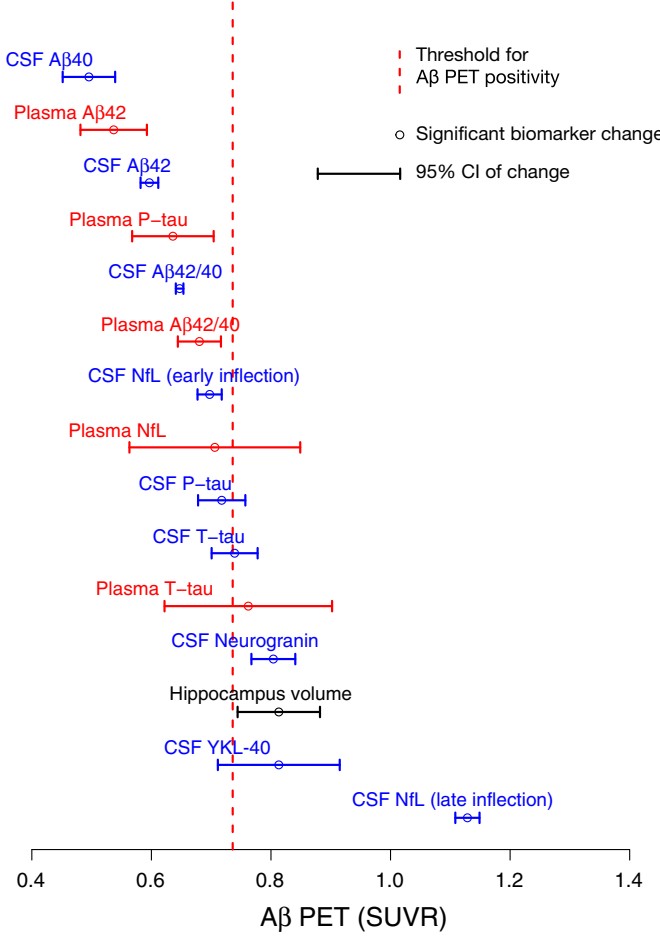

**Figure 2. Point of significant biomarker change with 95% CIs.**

Change points (also referred to as a significant biomarker change) with 95% CIs of the modeled biomarker trajectories are shown in Fig 1. Hippocampus volume divided by total intracranial volume was added for reference.

### Plasma biomarker change points

The plasma biomarkers changed approximately at the same point as the corresponding CSF biomarkers (Fig 2, Appendix Table S1), except for plasma neurogranin and Aβ40, which had non-significant models. Compared to CSF, all plasma biomarkers had wider 95% CIs, indicating greater variability in the early Aβ phase and/or less rapid biomarker changes.

### Comparisons of CSF assays from five different manufacturers

This comparative analysis was performed on a subset of the study population where complete data for all assays were available ($n = 352$ vs. $n = 377$ in the total population). Spline models for Aβ42 (Elecsys®, EUROIMMUN, INNOTEST, and MSD), Aβ42/40 (Elecsys®, EUROIMMUN, and MSD), P-tau (Elecsys®, EURO-IMMUN, INNOTEST, Lilly P-tau181, and Lilly P-tau217), and T-tau (Elecsys®, EUROIMMUN, and MSD) are shown in Fig 4A–D. Overall, the different biomarkers had very similar trajectories between

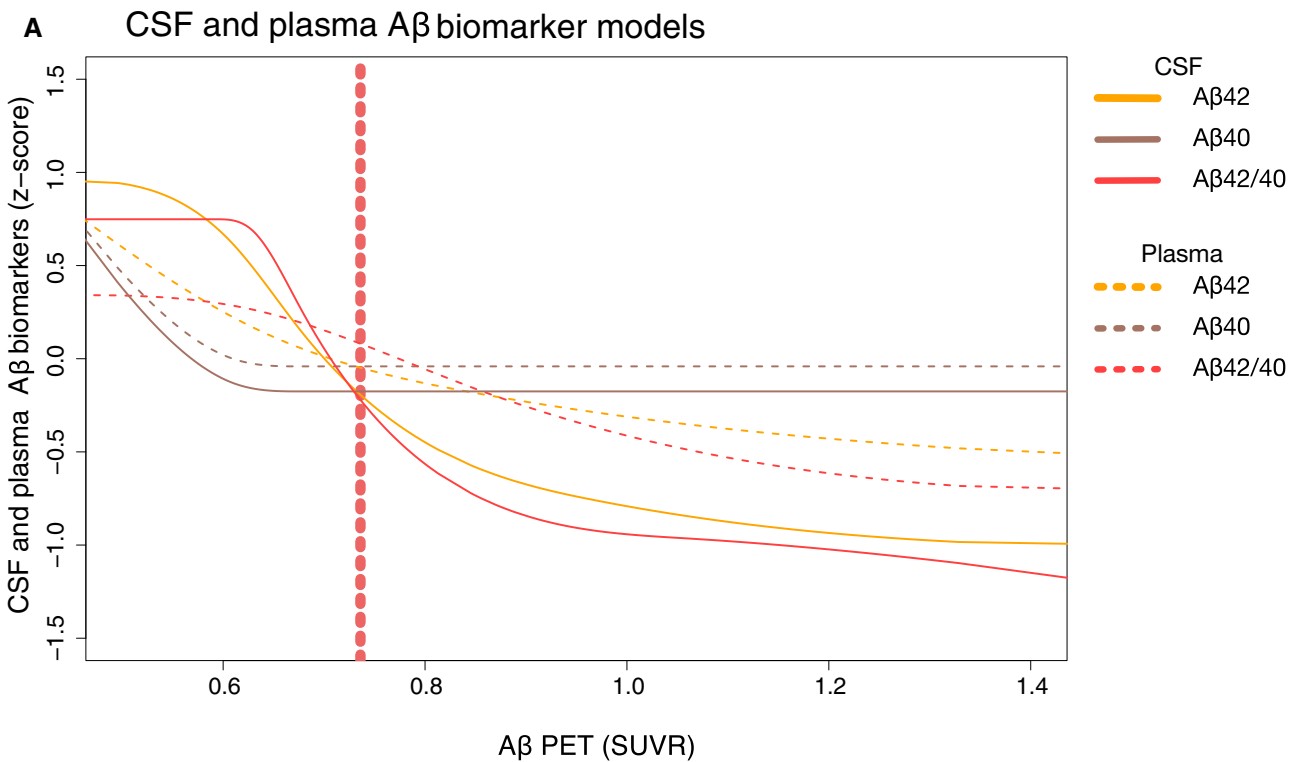

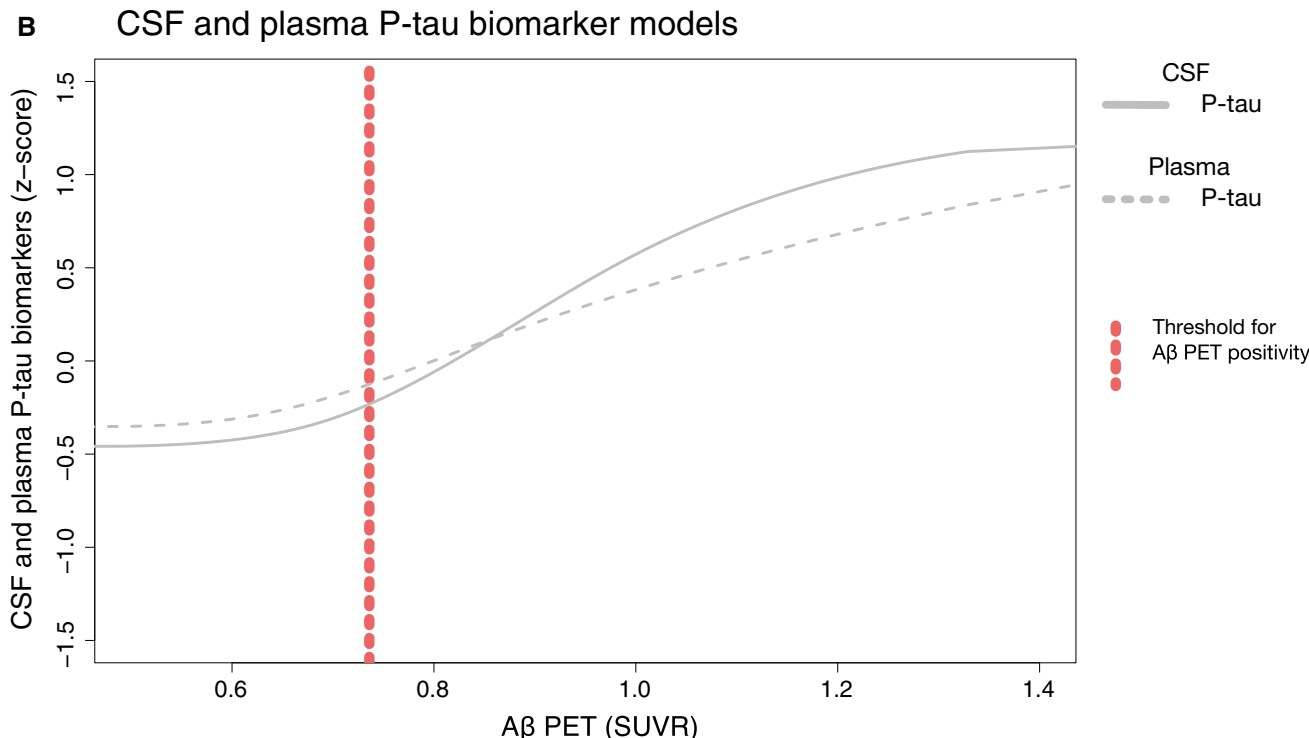

**Figure 3. Comparison of selected CSF and plasma biomarker models.**

A   Same models as in Fig 1A and B for CSF and plasma Aβ40, Aβ42, and Aβ42/40, but now in the same panel for easier comparison.

B   Spline models from the same dataset where there were no missing data for plasma P-tau (*n* = 343); i.e., the plasma P-tau curve is the same as in Fig 1B, but CSF P-tau is slightly different compared Fig 1A.

Data information: To facilitate comparisons between different CSF and plasma biomarkers, the levels have been transformed to *z*-scores based on the distribution in the present population (i.e., a *z*-score of 0 corresponds to the mean of the study cohort).

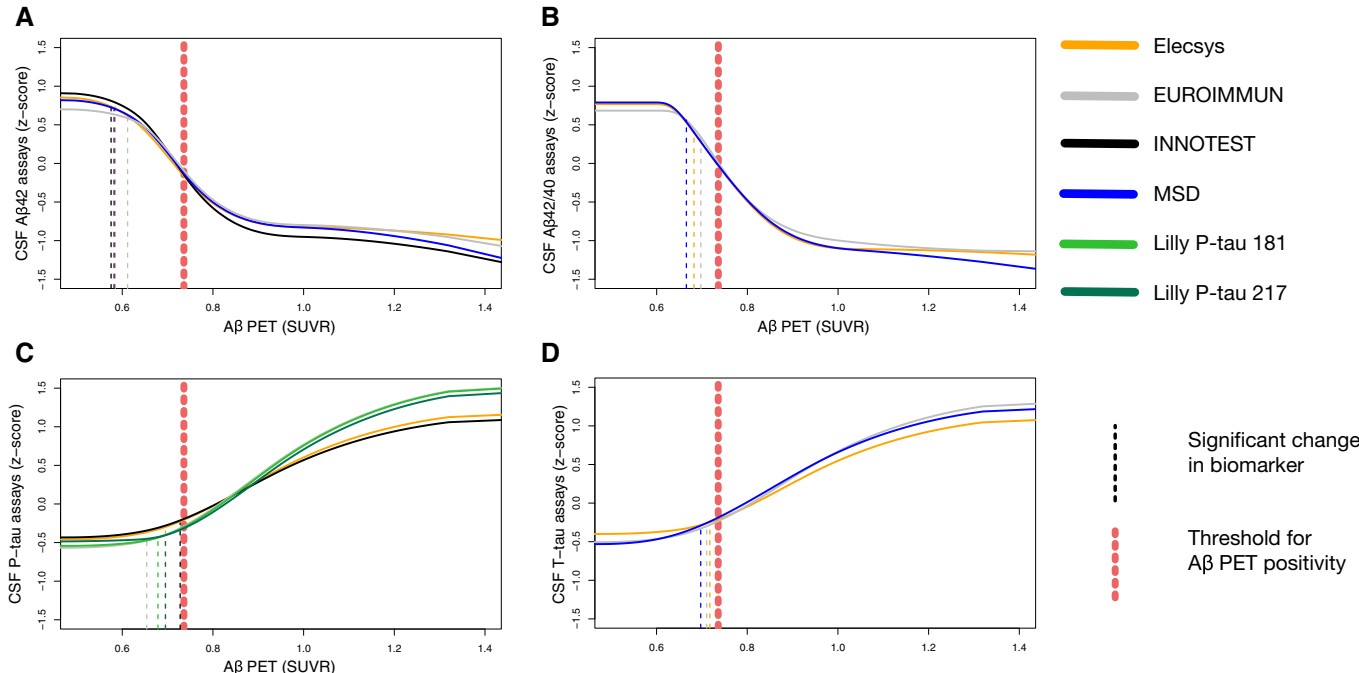

**Figure 4. Comparison of CSF biomarker trajectories from five different vendors.**

A–D The biomarker data were fitted using monotone spline models for CSF Aβ42 (A), Aβ42/40 (B), P-tau (C), and T-tau (D) assays, where Aβ PET SUVR acted as a proxy for time in AD. Point of change on the trajectory (also referred to as significant biomarker change) is shown as vertical dashed lines. Significant changes were identified for all biomarkers, but some were almost identical and are therefore partially hidden (see Fig 5 for a better overview of change points). Note that this analysis was performed on a slightly smaller sample where all participants had a complete dataset of all assays (*n* = 352 vs. *n* = 377 in the whole population).

assays. No significant differences were seen in change points between biomarker results obtained using any of the assays (Fig 5).

## Discussion

In this study of 377 individuals who were cognitively unimpaired (*n* = 242) or had MCI (*n* = 135), we examined 7 CSF and 6 plasma biomarkers in relation to fibrillar Aβ accumulation (measured using Aβ PET) to understand their evolution during the development of AD prior to the onset of dementia. In CSF, we found that the first significant changes were seen in Aβ42, followed closely by P-tau and T-tau (which all changed before Aβ PET positivity and concurrently with the Aβ42/Aβ40 ratio). Overt neurodegeneration (as measured by hippocampus volume and a second, more pronounced NfL increase) occurred later, after the threshold for Aβ positivity. In significantly modeled plasma biomarkers, we found no differences in change points compared with CSF (i.e., plasma biomarker changes were neither significantly earlier nor later than CSF; Fig 2). When comparing the trajectories of CSF and plasma biomarkers, we found that they were similar for Aβ42, Aβ40, Aβ42/40, and P-tau (Fig 3). Finally, we compared assays for CSF Aβ42, Aβ42/40, P-tau, and T-tau from different manufacturers and found that they were similar and could confirm the findings of the Elecsys® assays (Figs 4 and 5).

This is the first study to conduct a direct comparison of the CSF biomarkers Aβ42, Aβ40, T-tau, P-tau, neurogranin, YKL-40, and NfL during increasing Aβ accumulation, and also the first study to

include the corresponding plasma biomarker (except for YKL-40, which was not measured in plasma). Two recent studies have examined biomarker changes in the rare autosomal dominant variant of AD using estimated time to onset of cognitive symptoms as time variable in the Dominantly Inherited Alzheimer's Network (DIAN) disease study (McDade *et al*, 2018; Schindler *et al*, 2019). Despite using different samples and assays, they showed a similar sequence of change in biomarker levels with Aβ42 changing first, shortly after followed by P-tau, and then YKL-40 (neurogranin was also included in one of the studies but did not diverge significantly from healthy controls). A similar sequence of CSF Aβ42, T-tau, and P-tau changes has also recently been demonstrated in the Alzheimer's Disease Neuroimaging Initiative (ADNI) using longitudinal Aβ PET to estimate the time course of disease (Insel P *et al*, under review).

Overall, these findings agree with the current AD models postulating that Aβ is an initiating factor in the pathogenesis and that we hitherto do not have a biomarker that changes before Aβ (Jack *et al*, 2013). The significant change in CSF Aβ42 before the Aβ PET threshold for Aβ positivity (Fig 2) is in agreement with previous studies showing that CSF Aβ42 becomes abnormal prior to Aβ PET (Palmqvist *et al*, 2016, 2017; Vlassenko *et al*, 2016). Further, the later change in Aβ42/40 compared to Aβ42 might be one of the explanations for the higher agreement between (dichotomous) Aβ PET and Aβ42/40 compared with Aβ PET and Aβ42 (Fig 2), which has been reported previously (Janelidze *et al*, 2016, 2017; Lewczuk *et al*, 2017; Doecke *et al*, 2018). Although this temporal difference in Aβ42 and Aβ42/40 is in agreement with the spline model showing an initial drop in Aβ40 followed by stable levels (Fig 3A), the

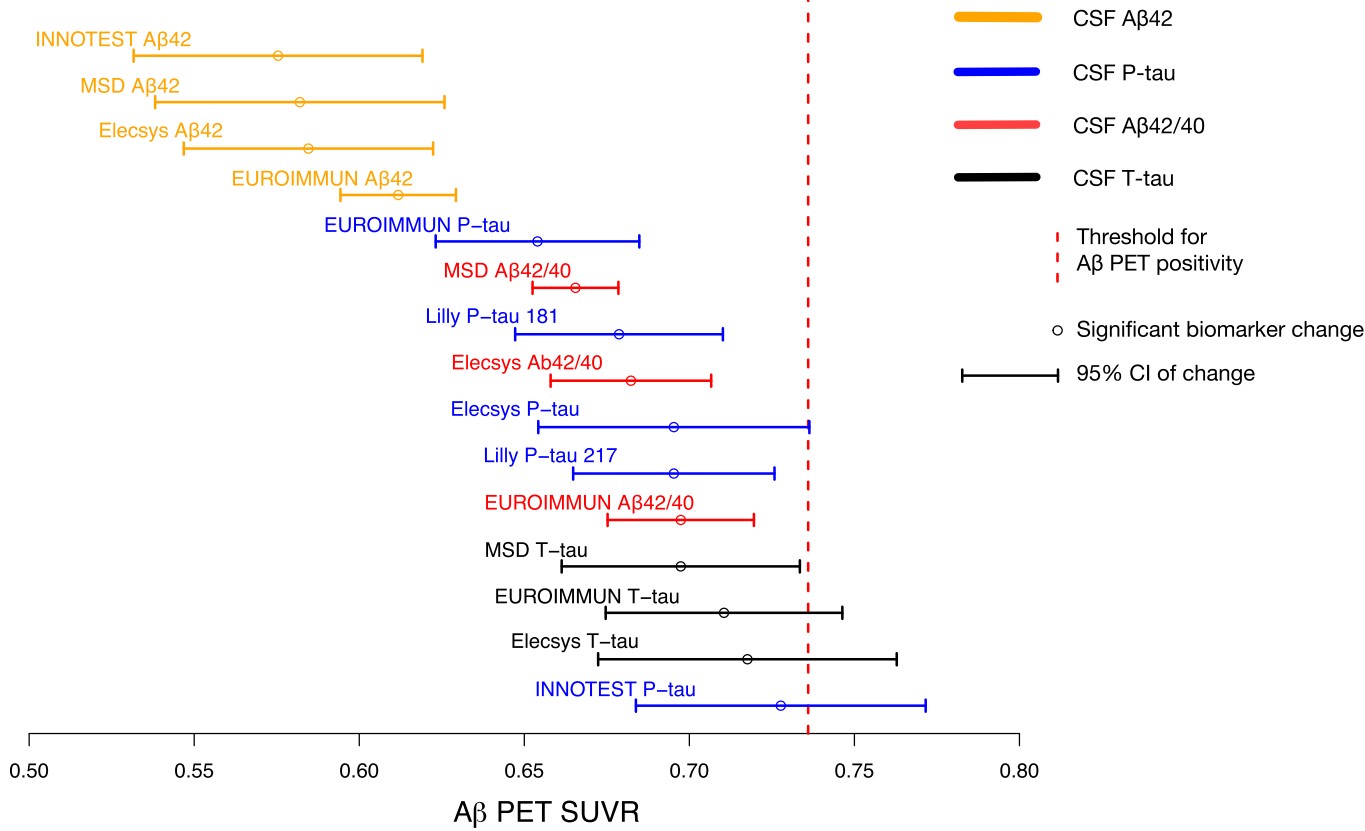

**Figure 5. Comparison of change points for the different Aβ42, Aβ42/40, T-tau, and P-tau assays.**

Significant biomarker changes with 95% CIs of the modeled biomarker trajectories are shown in Fig 4. Note that this analysis was performed on a smaller sample ($n = 352$ vs. $n = 377$) where all participants had a complete dataset of all assays, which gave slightly different results for the Elecsys® assay compared to the full dataset.

earlier change in Aβ42 is not supported by a recent study on autosomal dominant AD (Schindler *et al*, 2019). The Aβ40 finding should also be interpreted cautiously since this initial decline was driven by few individuals and the Aβ40 model was barely significant for CSF ($P = 0.02$; Appendix Fig S1) and was not significant for plasma ($P = 0.37$; Appendix Fig S2). Even though there are previous studies supporting the use of CSF Aβ42 measured in isolation to detect early Aβ accumulation (Mattsson *et al*, 2015, 2019b; Palmqvist *et al*, 2016, 2017), the ratio probably provides a more reliable measure of accumulating Aβ fibrils and increases its specificity since Aβ40 acts as a reference peptide that can, for example, account for inter-individual differences in CSF concentrations and differences in pre-analytical handling of the samples which otherwise may lead to false-positive or false-negative results using just Aβ42 (Janelidze *et al*, 2016; Lewczuk *et al*, 2017).

Although the biomarkers for tau phosphorylation state (P-tau), synaptic dysfunction (neurogranin), glial activation and inflammation (YKL-40), and neurodegeneration (NfL and T-tau) changed after CSF Aβ42, they still changed at a very early stage, just before or around the threshold for Aβ PET positivity. These results indicate that the known and measurable pathological mechanisms of AD all develop fairly simultaneously early on in the disease and that a long-lasting manifest Aβ stage ("Aβ positivity") is not required before alterations in other pathological mechanisms occur. This is

well in accordance with earlier work, showing that both brain hypometabolism and cognitive decline start to accelerate before Aβ positivity is reached (Insel *et al*, 2016, 2017). However, to complicate matters, one must consider that we are examining biomarkers and not actual pathological mechanisms. The validity (i.e., does the biomarker measure what it is supposed to measure) may thus clearly affect these results. It is, for example, possible that the early increase in P-tau reflects a neuronal reaction to the amyloidosis, rather than actual neurofibrillary tau pathology (Mattsson *et al*, 2017b; Sato *et al*, 2018), or that increased neuronal activity and secretion of a variety of intra-neuronal proteins could be an early AD event (Cirrito *et al*, 2008; Li *et al*, 2013). The latter case would result in an increase in several peptides in the extracellular space without any actual new pathological mechanisms taking place (other than increased secretion). This could for, example, also cause the initial increase in CSF NfL that is too subtle to be clinically relevant (early change point; Figs 1A and 2), followed by a more marked change better corresponding to overt neurodegeneration (late change point; Figs 1A and 2). This hypothesis is also in agreement with the contradictory studies showing that subtle longitudinal changes in NfL can be detected 16 years before the onset of cognitive symptoms (Preische *et al*, 2019), while cross-sectional differences are only noticeable at the MCI and dementia stage (Mattsson *et al*, 2019a,b).

In addition to the validity, the sensitivity of the biomarker (for detecting the underlying pathology) may also affect the results. For example, if CSF P-tau is much less sensitive to accumulating tau pathology in the brain than CSF Aβ42 to Aβ, we would find that CSF Aβ42 changed earlier even if tau was an earlier pathological mechanism in the brain. We therefore want to note that the identified order of biomarker changes (Fig 2) refers to the actual biomarkers, which may or may not translate to the order in which the underlying pathology appears in the brain.

When comparing the 95% CIs of the change points for CSF and corresponding plasma biomarkers, they overlapped (Fig 2; Appendix Table S1). Nonetheless, there were still considerable differences between their trajectories. Overall, the plasma biomarker models had considerably lower $r^2$ values than the corresponding CSF biomarkers and exhibited smaller dynamic ranges (Appendix Figs S1 and S2, and Figs 1A and B, and 3A). No changes were seen in plasma neurogranin indicating that it is not associated with Aβ-induced synaptic dysfunction. This was an expected result given the lack of correlation of CSF with plasma neurogranin concentrations, and that plasma neurogranin concentration probably reflects extra-cerebral expression of the protein (Kvartsberg et al, 2015). Plasma NfL and T-tau were barely significant in the models ($P = 0.02$ and $P = 0.04$, respectively; Appendix Fig S1). Since these latter two biomarkers are known to increase more markedly later in the disease (Mattsson et al, 2016a,b, 2017a,b), this finding can probably to some extent be explained by the fact that we only included non-demented individuals. In contrast to the above plasma biomarkers, plasma P-tau had a trajectory that was more alike the corresponding CSF biomarker (Fig 3B) and also had the highest $r^2$ value of all plasma biomarkers (Appendix Fig S2), which in the present study highlights it as the best plasma biomarker to track Aβ accumulation.

Regarding assays, we chose to use Elecsys® assays on a cobas e 601 instrument in the main analysis, since this was the only type of assay with available data for all standard CSF AD biomarkers (Aβ42, Aβ40, P-tau, and T-tau) and therefore allowed for a direct comparison of these biomarkers without confounding differences between manufacturers and techniques. However, the comparison of different vendors (Roche, MSD, EUROIMMUN, INNOTEST, and Lilly) showed no major differences in biomarkers trajectories (Fig 4) or significant differences in change points (Fig 5), indicating that these results are not assay/vendor specific. This comparative analysis also included a specific comparison between CSF P-tau181 and CSF P-tau217 (i.e., phosphorylated at the 181 and 217 position of threonine), but using the present statistics they were similar both in terms of trajectories (Fig 4C) and in terms of significant biomarker changes (Fig 5).

A shortcoming of our study design was that we only included cross-sectional CSF and plasma data. Previous studies have shown that earlier biomarker changes can be found using longitudinal, intra-individual changes in biomarkers (McDade et al, 2018; Preische et al, 2019), although this requires a good assay reliability and stringent CSF handling protocol over time. An additional shortcoming was the use of cross-sectional Aβ PET SUVR as a proxy of time in the disease. Although a gradual Aβ accumulation is seen throughout the development of AD (Villemagne et al, 2013), it is not certain that a higher SUVR always indicate a longer disease duration/later AD stage compared with a lower SUVR, since it is possible that different individuals have different starting levels of SUVR and different accumulation rates. Further, a non-linear accumulation rate of Aβ has been observed in AD (Villemagne et al, 2013). This does not affect the order of biomarker changes or whether the change occurred before or after Aβ positivity in the present study. However, one cannot, e.g., assume that the time between significant biomarker changes is exactly the same for CSF Aβ42 (0.60 SUVR) and CSF NfL (early change at 0.70 SUVR) as for CSF NfL (early change at 0.70 SUVR) and CSF neurogranin (0.80 SUVR) even though the differences are 0.1 SUVR for both (Appendix Table S1). These two shortcomings could in future studies to some extent be accounted for by including a follow-up Aβ PET and the Aβ rate as recently done in the ADNI study (Insel et al, 2017). The potential flaw of using cross-sectional SUVR as time variable in AD is probably most prominent among those individuals with very low SUVR, since none or very few of these individuals are on an AD trajectory. To account for this, we used monotone spline analyses that only allowed the biomarkers to change in one direction. This required us to have an a priori hypothesis about the direction of the biomarker (increase or decrease as SUVR increased), which thus minimized potential initial biomarker fluctuations before a clear, unidirectional biomarker change was seen. However, the monotonicity has the shortcoming that it may miss later, paradoxical biomarker changes as, e.g., reported in the DIAN study for CSF P-tau (McDade et al, 2018), although this does not seem to be the case when studying the actual data points for CSF P-tau in our study (Appendix Table S1).

The use of Aβ accumulation as a time variable could limit our ability to determine the point when a non-Aβ biomarker that is potentially earlier than Aβ accumulation change. However, since the non-Aβ biomarkers all started with flat trajectories and the first significant change was seen in Aβ42 (Fig 1A), this was likely not a major issue. We could thus confirm that using the presently available biomarkers, Aβ seems to be the initial disease mechanism in AD. This favors the ongoing and future trials targeting Aβ to halt AD. However, the closely accompanied increase in P-tau before Aβ positivity and also increases in markers for neuroinflammation, synaptic dysfunction, and neurodegeneration around the point of Aβ positivity suggest that these are also very early processes. Including Aβ-positive individuals using a standard Aβ PET cutoff in clinical trials might thus select a patient population where it may already be too late to get maximum treatment effect using a therapy that only targets Aβ, which is implicated by the failed trials (Knopman, 2019). A potential solution might thus be to enroll persons with subthreshold amyloidosis or to initiate a combination of therapies targeting different processes (Insel et al, 2017; Bischof & Jacobs, 2019).

# Materials and Methods

## Participants

The study population was included from the prospective Swedish BioFINDER study. The participants were included between 2010 and 2015 from the southern part of Sweden. All BioFINDER participants with complete [18]F-flutemetamol PET, CSF, MRI, and plasma data were included in the present study ($n = 377$), except for plasma

P-tau217 that had missing data for 34 individuals. Only cross-sectional CSF, plasma, MRI, and PET data were used. The participants were either cognitively healthy controls ($n = 148$) or patients with subjective cognitive decline (SCD; $n = 114$; controls and SCD were combined in a CU group) or MCI ($n = 147$; Jack et al, 2018).

The cognitively healthy controls in BioFINDER were enrolled from the Swedish sub-cohort of the population-based EPIC cohort (the Malmö Diet and Cancer study; Borland et al, 2017). The inclusion criteria were (i) absence of cognitive symptoms as assessed by a physician with special interest in cognitive disorders, (ii) age ≥ 60 years, (iii) MMSE score 28–30 at screening visit, (iv) do not fulfill the criteria for MCI or any dementia; and (v) speaks and understands Swedish to the extent that an interpreter was not necessary for the patient to fully understand the study information and cognitive tests. The exclusion criteria were (i) significant unstable systemic illness or organ failure, such as terminal cancer, that makes it difficult to participate in the study; (ii) current significant alcohol or substance misuse; (iii) refusing lumbar puncture or MRI; and (iv) significant neurological or psychiatric illness.

The inclusion criteria for patients with SCD or MCI were (i) referred to the memory clinic at Skåne University Hospital or Ängelholm Hospital in Sweden due to cognitive symptoms experienced by the patient and/or informant. These symptoms did not have to be memory complaints, but could also be executive, visuo-spatial, language, praxis, or psychomotor complaints; (ii) age between 60 and 80 years; (iii) MMSE score of 24–30 points at the baseline visit; (iv) do not fulfill the criteria for any dementia; and (v) speaks and understands Swedish to the extent that an interpreter was not necessary for the patient to fully understand the study information and neuropsychological tests. The exclusion criteria were (i) significant unstable systemic illness or organ failure, such as terminal cancer, that makes it difficult to participate in the study; (ii) current significant alcohol or substance misuse; (iii) refusing lumbar puncture or neuropsychological assessment; and (iv) the cognitive impairment at baseline visit can with certainty be explained by another condition or disease such as normal pressure hydrocephalus, major cerebral hemorrhage, brain infection, brain tumor, multiple sclerosis, epilepsy, psychotic disorders, severe depression, and alcohol abuse in the last 5 years, ongoing medication with drugs that invariably cause cognitive impairment (such as high-dose benzodiazepines).

The experiments conformed to the principles set out in the WMA Declaration of Helsinki and the Department of Health and Human Services Belmont Report. All participants gave written informed consent, and the study was approved by the regional ethics committee in Lund, Sweden. Further information about the study can be found at http://biofinder.se.

**Main CSF analysis**

LP and CSF handling followed a structured pre-analytical protocol (Palmqvist et al, 2014). The Elecsys® immunoassays (Roche Diagnostics) were used for the main analysis of CSF Aβ42, Aβ40, T-tau, and P-tau (Hansson et al, 2018). NfL and neurogranin were analyzed as previously described (Mattsson et al, 2016a,b; Hansson et al, 2017; Kvartsberg et al, 2019). YKL-40 concentrations were measured using ELISA kits according to the manufacturer's recommendations (YKL-40 R&D Systems, Inc., Minneapolis, MN, USA)

(Janelidze et al, 2018). Elecsys® biomarkers, NfL, neurogranin, and YKL-40 were analyzed at the Clinical Neurochemistry Laboratory, University of Gothenburg, Mölndal, Sweden.

**Plasma analysis**

Blood samples were collected at the same time as CSF samples and handled according to a structured pre-analytical protocol (Palmqvist et al, 2019). Plasma Aβ42, Aβ40, and T-tau concentrations were measured using the Elecsys® immunoassays on a cobas e® 601 analyzer (Palmqvist et al, 2019). NfL was analyzed as previously described (Hansson et al, 2017). Neurogranin concentration was measured using ELISA kits according to the manufacturer's recommendations (Euroimmun AG, Lübeck, Germany). P-tau217 concentrations were measured using the Lilly assay as described below. Note that plasma P-tau217 was missing for 34 cases. Plasma P-tau217 concentration was measured at Lilly Research Laboratories, IN, USA, and neurogranin at Euroimmun AG, Lübeck, Germany. All other plasma samples were analyzed at the Clinical Neurochemistry Laboratory, University of Gothenburg, Mölndal, Sweden.

*Plasma analysis of Lilly P-tau217*
The assay was performed on a streptavidin small-spot plate using the MSD platform. Biotinylated-IBA493 was used as a capture antibody (anti-P-tau217) and SULFO-TAG-4G10-E2 (anti-Tau) for the detector. Antibodies were conjugated with Sulfo-NHS-Biotin (Thermo Scientific, catalog number: 21329) or MSD GOLD SULFO-TAG NHS-Ester (MSD, catalog number: R91AO-1) according to the manufacturer's protocol. The assay was calibrated using a recombinant tau (4R2N, NCBI tau v2) protein that was phosphorylated in vitro using a reaction with glycogen synthase kinase-3 and characterized by mass spectrometry. The sample was thawed on wet ice, briefly vortexed, and centrifuged at 2,000 $g$ for 10 min, and plasma was diluted 1:2.5 in sample buffer (50 mM HEPES, 300 mM NaCl, 5 mM EDTA, 5 mM EGTA, 1% Triton X-100, 1% MSD blocker A, 2% PEG) with the addition of heterophilic blocking reagent 1 to a concentration of 200 μg/ml (Scantibodies Inc, catalog number: 3KC533). Calibrator diluent was identical to sample diluent. In order to run the assay, MSD small-spot streptavidin (MSD, L45SA)-coated plates were blocked for 1 h at room temperature with 200 μl of 3% BSA in DPBS with 650 rpm shaking on a plate shaker. The plates were then washed three times with 200 μl of wash buffer (PBS + 0.05% Tween 20), and 25 μl of biotinylated-IBA493 capture antibody at 0.5 μg/ml (diluted in DPS + 0.1% BSA + 0.05% Tween 20 + 2% PEG) was added for the P-tau217 plates and incubated for 1 h at room temperature with 650 rpm shaking on a plate shaker. The plates were again washed three times with 200 μl of wash buffer, and 50 μl of diluted calibrator or sample was added to the plate and incubated for 2 h at room temperature with 650 rpm shaking on a plate shaker. The plates were then washed three times with 200 μl of wash buffer, and 25 μl of SULFO-tagged E2 detection antibody was added at 0.05 μg/ml (diluted in MSD Diluent 35 + 2% PEG) for P-tau217 plates and incubated for 1 h at room temperature with 650 rpm shaking on a plate shaker. The plates were washed a final time with 200 μl of wash buffer, and 150 μl of 2× MSD Read Buffer T with Surfactant (MSD, R92TC) was added to each plate and read on the MSD SQ120 within 10 min of read buffer addition.

## Comparison of CSF assays from different manufacturers

In a secondary analysis, we compared the performance of the following five immunoassays: Elecsys® (Aβ42, Aβ40, T-tau, and P-tau; Roche Diagnostics, Penzberg, Germany), EUROIMMUN (Aβ42, Aβ40, T-tau, and P-tau; Euroimmun AG, Lübeck, Germany), INNOTEST (Aβ42 and P-tau; Fujirebio, Gent, Belgium), MSD (Aβ42, Aβ40, T-tau; Meso Scale Discovery, Rockville, MD, USA), and Lilly (P-tau181 and P-tau217 (i.e., tau phosphorylated at a threonine amino acid at residue 181 and 217, respectively); Lilly Research Laboratories, IN, USA). Note that the Lilly P-tau217 assay was the only one targeting the 217 site, all other P-tau assays targeted 181 (Elecsys P-tau, EURO-IMMUN P-tau, INNOTEST P-tau, and Lilly P-tau181). The Lilly P-tau assays are described below. All other assays were performed according to the manufacturer's recommendations. Twenty-five participants had missing data for one or more assays, and 352 participants were thus included in this comparative analysis.

### CSF analysis of Lilly P-tau181 and P-tau217

The CSF Lilly P-tau181 and P-tau217 assays were designed for CSF analysis. Both assays were performed on a streptavidin small-spot plate using the Meso Scale Discovery (MSD) platform (Meso Scale Discovery, Rockville, MD, USA). Anti-P-tau217 antibody IBA413 was used as a capture antibody in the P-tau217 assay, whereas anti-P-tau181 antibody AT270 was used as a capture antibody in the P-tau181 assay. Both assays used a tau-specific antibody (LRL) as the detector. Antibodies were conjugated with Biotin (Thermo Scientific, catalog number: 21329) or SULFO-TAG (MSD, catalog number: R91AO-1). The assays were calibrated using a recombinant tau (4R2N) protein that was phosphorylated *in vitro* using a reaction with glycogen synthase kinase-3 and characterized by mass spectrometry. The samples were thawed on wet ice, briefly vortexed, and diluted 1:8 in Diluent 35 (MSD, catalog number: R50AE) with the addition of a heterophilic blocking reagent to a concentration of 200 μg/ml (Scantibodies Inc, catalog number: 3KC533). In order to perform the assays, MSD small-spot streptavidin-coated plates (MSD, catalog number: L45SA) were blocked for 1 h at room temperature with 200 μl of 3% BSA in DPBS with 650 rpm shaking on a plate shaker. The plates were then washed three times with 200 μl of wash buffer (PBS + 0.05% Tween 20), and 25 μl of biotinylated capture antibody (AT270 for P-tau181 or IBA413 for P-tau217) at 1 and 0.1 μg/ml, respectively, were added to the wells and incubated for 1 h at room temperature with 650 rpm shaking on a plate shaker. The plates were again washed three times with 200 μl of wash buffer, and 50 μl of diluted calibrator or sample was added to each well and incubated for 2 h at room temperature with 650 rpm shaking on a plate shaker. The plates were then washed three times with 200 μl of wash buffer, and 25 μl of SULFO-tagged LRL detection antibody was added at 3 μg/ml for the P-tau181 and at 0.5 μg/ml for the P-tau217 plates and incubated for 1 h at room temperature with 650 rpm shaking on a plate shaker. The plates were washed a final time with 200 μl of wash buffer, and 150 μl of 2× MSD Read Buffer T with Surfactant (MSD, catalog number: R92TC) was added to each plate and read on the MSD SQ120 within 10 min of read buffer addition. Samples were analyzed in duplicates, and the mean of duplicates was used in statistical analysis. The assay performances across all P-tau217 and P-tau181 plates are summarized in Appendix Table S2 with a 10 and 50 pg/ml QC buffer spike and high, medium, and low control CSF samples. Individual measurements all fell within 20% of the mean for QC and control samples.

### Aβ PET

[18]F-Flutemetamol was used for Aβ PET imaging. We extracted the average standardized uptake value ratio (SUVR) from brain regions prone to early Aβ accumulation (Palmqvist *et al*, 2017), relative the uptake in a previously published composite reference region (Landau *et al*, 2015). The Aβ PET uptake (SUVR) in this region was used as a proxy for "time in disease", based on the assumption that the development of fibrillar Aβ deposition is the key defining event of AD (Hardy & Selkoe, 2002) and continues to accumulate during the disease progression (Villemagne *et al*, 2013). The early Aβ accumulating region was used instead of a large neocortical region to increase the sensitivity to detect early CSF and plasma biomarker changes. This specific region of interest (ROI) was comprised predominantly of the posterior cingulate cortex and precuneus, the subgenual part of the anterior cingulate cortex and smaller parts of the angular gyrus, posterior middle temporal gyrus, and middle frontal gyrus (Hahn *et al*, 2019). Aβ status (positive or negative) was established based on mixture modeling statistics (Benaglia *et al*, 2009) in the present study population using the SUVR data from the early Aβ accumulating ROI (> 0.736 SUVR defined Aβ positivity). Since it was derived from the present population, it has not been validated against neuropathological data. However, this method of establishing Aβ cutoffs has successfully been used in many previous publications (Villeneuve *et al*, 2015; Palmqvist *et al*, 2016; Bertens *et al*, 2017).

### MRI

All participants were examined using the same MR scanner (3 Tesla Siemens Tim Trio). The MR scanning and imaging procedures have been described previously (Palmqvist *et al*, 2017). FreeSurfer software (version 5.3) was used to extract data for the total intracranial volume and total hippocampal volume (left and right).

### Statistical analysis

The Mann–Whitney test was used for group comparisons (Table 1). Aβ PET SUVR was used as a proxy for progression of AD (*x*-axis in the figures). The relationship between each biomarker response and the Aβ PET uptake was modeled using monotone penalized regression splines. Generalized cross-validation was used to tune the smoothing parameter (Wood, 1994). Plasma NfL, P-tau, and T-tau as well as CSF YKL-40 and NfL had skewed distributions (based on visual inspection of histograms) and were therefore natural log-transformed. This resulted in normal distributions for these biomarkers (but one single outlier was observed for plasma NfL). All CSF and plasma biomarkers were transformed to *z*-scores (based on the distribution in the present population) to facilitate comparison of results between biomarkers. Change points in the biomarker splines were established via a bootstrap-estimated standard error (SE) of the initial point (lowest SUVR, i.e., had no signs of Aβ

accumulation on Aβ PET) of the curve, using 500 bootstrap samples. A significant biomarker change (i.e., change point of the curve) was defined as a change in 2 SE from the first point of the original biomarker curve from the whole dataset, similar to a previous publication (Insel *et al*, 2017). To compare differences between the significant change points of the biomarkers, 95% CIs were created (original significant change point $\pm$ 1.96 SE from a new set of 500 bootstrapped samples). CSF NfL had a sigmoid spline (a subtle initial increase was followed by a plateau, and a second manifest increase). Therefore, two change points were determined for CSF NfL: one early and one late change. The middle of the plateau was used as the second point of biomarker normality (at 0.949 SUVR). Change points were only calculated for significant models (i.e., not for plasma Aβ40 and plasma neurogranin). R version 3.4.3 was used for all statistical analyses.

## Data availability

The study data are not publicly available for download, but might be retrieved from the principal investigator professor Oskar Hansson.

**Expanded View** for this article is available online.

### Acknowledgements
Work at the authors' research center was supported by the European Research Council, the Swedish Research Council, the Knut and Alice Wallenberg foundation, the Marianne and Marcus Wallenberg foundation, the Strategic Research Area MultiPark (Multidisciplinary Research in Parkinson's disease) at Lund University, the Swedish Alzheimer Association, the Swedish Brain Foundation, The Parkinson foundation of Sweden, The Parkinson Research Foundation, the Skåne University Hospital Foundation, and the Swedish federal government under the ALF agreement.

### Author contributions
Concept and design: SP, PSI, NM, and OH. Statistical Analysis: SP, PSI, and NM. Acquisition of data, biomarker analysis or interpretation of data: SP, SJ, ES, HZ, UE, NM, KB, JLD, XC, BB, and OH. Drafting of the manuscript: SP. Critical revision of the manuscript for important intellectual content: PSI, ES, HZ, NM, SJ, UE, KB, JLD, XC, BB, and OH. Obtained funding: SP, NM, and OH. Supervision: OH.

### Conflict of interest
SP, PSI, SJ, ES, and NM report no disclosures. UE is a current employee of the Roche Group. BB is a current employee of EUROIMMUN. JLD and XC are current employees of Lilly. KB has served as a consultant or at advisory boards for Alzheon, BioArctic, Biogen, Eli Lilly, Fujirebio Europe, IBL International, Merck, Novartis, Pfizer, and Roche Diagnostics; is a co-founder of Brain Biomarker Solutions in Gothenburg AB, a GU Ventures-based platform company at the University of Gothenburg; and has received research support (for the institution) from Roche Diagnostics and Fujirebio Europe. HZ has served at scientific advisory boards for Roche Diagnostics, Eli Lilly, and Wave; has received travel support from Teva; and is a co-founder of Brain Biomarker Solutions in Gothenburg AB, a GU Ventures-based platform company at the University of Gothenburg. OH has acquired research support (for the institution) from Roche, GE Healthcare, Biogen, AVID Radiopharmaceuticals, Fujirebio, and Euroimmun. In the past 2 years, he has received consultancy/speaker fees (paid to the institution) from Lilly, Roche, and Fujirebio.

### The paper explained

#### Problem
Continued failures in clinical trials for Alzheimer's disease (AD) against presumably the correct drug targets highlight the need for further research to understand important disease mechanisms and at what stage they occur and become measurable.

#### Results
The study examines seven cerebrospinal fluid (CSF) and six plasma biomarkers in relation to increasing β-amyloid (Aβ) PET uptake to understand their evolution during the development of AD. The first changes were seen in Aβ biomarkers, closely followed by soluble tau, and then approximately simultaneously in markers of neuroinflammation, synaptic dysfunction, and neurodegeneration. The plasma biomarkers for Aβ, tau, and neurodegeneration started to change approximately at the same point as the corresponding CSF biomarkers. Especially, phosphorylated-tau (P-tau) in plasma behaved very similar to CSF P-tau. The main results were replicated using five different CSF assays for Aβ42, Aβ40, P-tau, and T-tau.

#### Impact
The study lends *in vivo* support to the amyloid cascade hypothesis in AD. It also shows that potential downstream mechanisms such as tau, neuroinflammation, and neurodegeneration are very early mechanisms that occur before or around the threshold for abnormal amyloid deposition ("amyloid positivity"), which may be important to consider when designing clinical AD trials. Finally, the study highlights the usefulness of plasma biomarkers in AD, especially plasma P-tau.

## For more information
www.biofinder.se

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
