## [Review Process File · EMBO Molecular Medicine]

Cerebrospinal fluid and plasma biomarker trajectories with increasing amyloid deposition in Alzheimer's disease

Sebastian Palmqvist, Philip S. Insel, Erik Stomrud, Shorena Janelidze, Henrik Zetterberg, Britta Brix, Udo Eichenlaub, Jeffrey L. Dage, Xiyun Chai, Kaj Blennow, Niklas Mattsson, Oskar Hansson

Review timeline:	Submission date:	16 July 2019
	Editorial Decision:	16 August 2019
	Revision received:	15 September 2019
	Editorial Decision:	2 October 2019
	Revision received:	9 October 2019
	Accepted:	14 October 2019

Editor: Céline Carret

Transaction Report:

1st Editorial Decision

16 August 2019

Thank you for the submission of your manuscript to EMBO Molecular Medicine. We have now heard back from the two referees whom we asked to evaluate your manuscript.

You will see that both referees are overall supportive of publication, pending further explanations, clarifications, discussions but also altering the title to better reflect the data. However, ref. 2 suggested not using "amyloid accumulation" nor "development" as these imply a longitudinal study.

We would therefore welcome the submission of a revised version within three months for further consideration and would like to encourage you to address all the criticisms raised as suggested to improve conclusiveness and clarity. Please note that EMBO Molecular Medicine strongly supports a single round of revision and that, as acceptance or rejection of the manuscript will depend on another round of review, your responses should be as complete as possible.

Please also contact us as soon as possible if similar work is published elsewhere. If other work is published, we may not be able to extend the revision period beyond three months.

I look forward to receiving your revised manuscript.

***** Reviewer's comments *****

Referee #1 (Comments on Novelty/Model System for Author):

The model is human which is great. The findings are important. The authors build a timecourse model with some assumptions about amyloid PET results. They discuss the caveats to this model, but they should probably change the title as there are limitations to the approach they use using amyloid PET results at a single point in building their time model from cross sectional data.

Referee #1 (Remarks for Author):

In this manuscript, the authors assess cerebrospinal fluid and plasma biomarker trajectories using amyloid PET as a proxy for disease progression. This study confirms the temporal order of fluid biomarker changes (e.g. A β 42 before neurodegeneration/neuronal injury/inflammatory markers) in concordance with the amyloid hypothesis and in agreement with previous studies, both longitudinal and cross-sectional. Rather than demonstrating novel findings, the authors use a novel method of analysis in A β PET SUVR is used as the "time" variable in these analyses. The current study found that using amyloid PET SUVR at a time variable in their model, that CSF A β 42 changed prior to A β PET positivity, followed by the A β 42/A β 40 ratio, P-tau and T-tau, and that Neurogranin, NfL, and YKL-40 did not change until after A β PET positivity. In addition to these main findings, a head-to-head assessment of fluid biomarker platforms/assays including, MSD, and EUROIMMUN and Lilly ELISAs showed no major difference to the Roche Elecsys platform used for the main analyses. Overall, this paper is well written and covers a well-thought-out study.

There are few issues for the authors to address:

Major issues

1. The main worry is the usage of A β PET as a proxy of disease state. The authors do state this as an additional shortcoming (page 12, second paragraph), but there is potential that the findings really pertain more to the relationship between PET SUVR and fluid biomarkers, than AD development and fluid biomarkers.
 - a. If the authors state that it is not certain whether a higher SUVR always indicates a longer disease duration or a later AD stage, then the title "Cerebrospinal fluid and plasma biomarker trajectories during the development of Alzheimer's disease" might be overstated. Perhaps "during Amyloid accumulation" would be more appropriate.
 - b. Prior to larger studies collecting enough longitudinal data to publish, the consensus was mainly that biomarkers of neurodegeneration/neuronal injury continued to increase throughout the stages of AD. However, more recent studies from DIAN and ADNI (Fagan et al., 2014, McDade et al., 2018, Sutphen et al., 2018) are starting to show that longitudinal decreases might feature prominently in the later stages of disease. In this context, equating a potentially non-linear biomarker (A β PET) as, a linear proxy of disease progress/time, and further using a monotone spline analysis to force biomarker change in only one direction may support conclusions that are not as nuanced as the complexity of AD requires - particularly in earlier disease stages.

Referee #2 (Remarks for Author):

Based on cross-sectional data Palmqvist et al describe a model of longitudinal change of a set of CSF and plasma biomarkers in a cohort of cognitively healthy individuals and MCI patients, in comparison to an amyloid PET cut-off. They report the timing of the inflection point compared to this cut-off (figures 1-3).

The sample size is appropriate, the statistical approach is sound, the results straightforward, the discussion balanced. The paper is well-written, and the data presented in a clear manner. The model of change in plasma biomarkers is novel. In particular the p tau plasma biomarker performed remarkably well and constituted one of the few plasma markers with an inflection point significantly preceding the timepoint of PET positivity. The comparison between assays of the same marker from different manufacturers (Figs 4-5) also adds to the novelty of the paper. The CSF results confirm what one would expect based on prior studies.

Main comment

1. The authors interpret the data as favoring the amyloid cascade hypothesis. This goes beyond what the data allow one to conclude: The comparator is amyloid PET which may bias the conclusions in favor of amyloid as a prime mover, furthermore the measures p-tau and total tau may have a low sensitivity for detecting early brain tau pathology and sensitivity may differ between amyloid and tau fluid biomarkers.
2. The fact that the longitudinal model is entirely based on cross-sectional data should be mentioned also in the methods and results.

3. Given the novelty of the plasma biomarker results, it would be of interest to add how accurately each of the plasma markers discriminates between amyloid PET-positive and amyloid PET-negative individuals.
4. The authors should mention explicitly on which dataset the mixture modelling was applied that led to the amyloid PET cut-off. According to the Hahn et al paper it is based on the broader BioFinder dataset. It is important to mention how many healthy controls and how many MCI or AD dementia patients this included as a mixture model will give different cut-offs for different cohorts. The authors should also mention whether this cut-off has been validated against a neuropathological standard-of-truth. Although the lack of validation of the amyloid PET cut-off against a gold standard is an issue, a shift in the amyloid cut-off would by no means alter the basic conclusions about the relative longitudinal change between the fluid biomarkers.
5. Supplementary figures: For some CSF assays (CSF A β 40, CSF YKL40, CSF NFL) and some plasma biomarker assays (plasma Abeta42, plasma Abeta40, NFL and t-tau), the r^2 is very low (0.02-0.08). Even when the p value is significant, for those assays with such a low r^2 the spline model in the main paper may be misleading and it would be better to omit these from the main figures because of the very low amount of variance explained by the model for these assays. Alternatively, it would be fair to highlight in the results that r^2 was very low for these.

Minor comments

1. On p 17 'to a lesser extent' should be expressed in mathematical terms, e.g. whether a weighting was applied.
2. Figures 1 & 3: Add to the legend that the Z scores are derived from the study cohort itself so that $Z = 0$ corresponds to the mean of the study cohort, rather than the mean of the norm group as would be more commonly the case. For correct interpretation of figure 3, the S.D. for the different tests in this cohort would be needed.
3. The value of the early change in A β 40 and A β 42 measured in isolation should be qualified as these measures have a relatively low specificity.
4. In the online supplement describing CSF and plasma ptau measurement, the authors should specify for CSF that this pertains to the EliLilly assay. For CSF, the authors describe antibodies against two phosphorylation sites, it is not clear which of the two is used in the further analyses. For the sake of consistency, on p 16 also mention which phosphorylation sites are targeted for the ptau assays from the other manufacturers. Suppl table 1 lists some of the performance parameters of the two CSF ptau assays but this should also be provided for the plasma ptau assay.
5. Several tests had a skewed distribution. The authors should mention which statistical test they used to determine the normality of the distribution and whether the distribution was then normal following log-transformation. If the data had a bimodal distribution, then log-transformation would probably not be appropriate.

1st Revision - authors' response

15 September 2019

Editor's Comments

You will see that both referees are overall supportive of publication, pending further explanations, clarifications, discussions but also altering the title to better reflect the data. However, ref. 2 suggested not using "amyloid accumulation" nor "development" as these imply a longitudinal study.

Author's reply:

All comments have been addressed in point-to-point replies below. We have now changed the title and removed both "amyloid accumulation" and "development". The new title is: "Cerebrospinal fluid and plasma biomarker trajectories with increasing amyloid deposition in Alzheimer's disease"

Reviewer #1 (Comments on Novelty/Model System for Author):

The model is human which is great. The findings are important. The authors build a timecourse model with some assumptions about amyloid PET results. They discuss the caveats to this model, but they should probably change the title as there are limitations to the approach, they use using amyloid PET results at a single point in building their time model from cross sectional data.

Author's reply:

We thank you for this comment. The title has been changed to: “Cerebrospinal fluid and plasma biomarker trajectories with increasing amyloid deposition in Alzheimer’s disease”. See further replies to referee 2, comment 1 regarding the title.

In this manuscript, the authors assess cerebrospinal fluid and plasma biomarker trajectories using amyloid PET as a proxy for disease progression. This study confirms the temporal order of fluid biomarker changes (e.g. A β 42 before neurodegeneration/neuronal injury/inflammatory markers) in concordance with the amyloid hypothesis and in agreement with previous studies, both longitudinal and cross-sectional. Rather than demonstrating novel findings, the authors use a novel method of analysis in A β PET SUVR is used as the "time" variable in these analyses. The current study found that using amyloid PET SUVR at a time variable in their model, that CSF A β 42 changed prior to A β PET positivity, followed by the A β 42/A β 40 ratio, P-tau and T-tau, and that Neurogranin, NfL, and YKL-40 did not change until after A β PET positivity. In addition to these main findings, a head-to-head assessment of fluid biomarker platforms/assays including, MSD, and EUROIMMUN and Lilly ELISAs showed no major difference to the Roche Elecsys platform used for the main analyses. Overall, this paper is well written and covers a well-thought-out study. There are few issues for the authors to address:

The main worry is the usage of A β PET as a proxy of disease state. The authors do state this as an additional shortcoming (page 12, second paragraph), but there is potential that the findings really pertain more to the relationship between PET SUVR and fluid biomarkers, than AD development and fluid biomarkers.

1a. If the authors state that it is not certain whether a higher SUVR always indicates a longer disease duration or a later AD stage, then the title "Cerebrospinal fluid and plasma biomarker trajectories during the development of Alzheimer's disease" might be overstated. Perhaps "during Amyloid accumulation" would be more appropriate.

Author’s reply:

Thank you for the comment. The editor and referee 2 have also commented on our title and suggested neither using “accumulation” nor “development” in the title. We have therefore changed it to “Cerebrospinal fluid and plasma biomarker trajectories with increasing amyloid deposition in Alzheimer’s disease”. Removing “Alzheimer’s disease” might make it difficult for the readers (of such a broad journal as EMBO Molecular Medicine) to understand in which field the paper is relevant. Also, increasing amyloid deposition is a hallmark of AD, which we believe warrants its use in the title (see e.g. Jack & Vermuri, Nat Rev Neurol, 2018).

1b. Prior to larger studies collecting enough longitudinal data to publish, the consensus was mainly that biomarkers of neurodegeneration/neuronal injury continued to increase throughout the stages of AD. However, more recent studies from DIAN and ADNI (Fagan et al., 2014, McDade et al., 2018, Sutphen et al., 2018) are starting to show that longitudinal decreases might feature prominently in the later stages of disease. In this context, equating a potentially non-linear biomarker (A β PET) as, a linear proxy of disease progress/time, and further using a monotone spline analysis to force biomarker change in only one direction may support conclusions that are not as nuanced as the complexity of AD requires - particularly in earlier disease stages.

Author’s reply:

We agree that the use of monotonicity in the models, although providing a more reliable measure for when the inflection point occurs as well as guard against edge effects, may have disadvantages especially if the biomarker trajectory goes in the other direction at a later stage. However, we only included non-demented individuals and paradoxical changes in the demented would thus not be seen in the present study. Also, we provide individual data in Supplementary Tables 1-2 and here we don’t see a trend for a paradoxical decrease/increase in any biomarker. Nonetheless, we have added this as a shortcoming (p. 13, last sentence of 1st paragraph):

“However, the monotonicity has the shortcoming that it may miss later, paradoxical biomarker changes as e.g. reported in the DIAN study for CSF P-tau (McDade et al., 2018), although this does not seem to be the case when studying the actual data points for CSF P-tau in our study (Supplementary Table 1).”

We agree that the non-linearity of amyloid accumulation (e.g. measured with amyloid PET) may have an effect if a reader tries to extrapolate differences between SUVR points to exact time differences (we refer to the order of inflection points, which is not affected by the non-linearity). To clarify this, we have added the following to the discussion (p. 12-13, last-1st paragraph):

“Further, a non-linear accumulation rate of A β has been observed in AD (Villemagne et al., 2013). This does not affect the order of biomarker changes or whether the change occurred before or after A β positivity in the present study. However, one cannot e.g. assume that the time between significant biomarker changes are exactly the same for CSF A β 42 (0.60 SUVR) and CSF NfL (early change at 0.70 SUVR) as for CSF NfL (early change at 0.70 SUVR) and CSF neurogranin (at 0.80 SUVR) even though the differences are approximately 0.1 SUVR for both (Supplementary Table 1).”

Referee #2 (Remarks for Author):

Based on cross-sectional data Palmqvist et al describe a model of longitudinal change of a set of CSF and plasma biomarkers in a cohort of cognitively healthy individuals and MCI patients, in comparison to an amyloid PET cut-off. They report the timing of the inflection point compared to this cut-off (figures 1-3).

The sample size is appropriate, the statistical approach is sound, the results straightforward, the discussion balanced. The paper is well-written, and the data presented in a clear manner. The model of change in plasma biomarkers is novel. In particular the p tau plasma biomarker performed remarkably well and constituted one of the few plasma markers with an inflection point significantly preceding the timepoint of PET positivity. The comparison between assays of the same marker from different manufacturers (Figs 4-5) also adds to the novelty of the paper. The CSF results confirm what one would expect based on prior studies.

Authors' reply:

We thank the reviewer the comment.

Main comment

1. The authors interpret the data as favoring the amyloid cascade hypothesis. This goes beyond what the data allow one to conclude: The comparator is amyloid PET which may bias the conclusions in favor of amyloid as a prime mover, furthermore the measures p-tau and total tau may have a low sensitivity for detecting early brain tau pathology and sensitivity may differ between amyloid and tau fluid biomarkers.

Authors' reply:

We agree that using amyloid PET as our time variable removes the possibility of detecting inflection points / starting points of biomarker changes that occur before amyloid starts to accumulate. However, such early non-amyloid biomarkers should already show a rate of change from the beginning (lowest SUVR) in Fig 1, which was not observed. We have addressed this limitation on p. 13, 2nd paragraph:

“The use of A β accumulation as a time variable could limit our ability to determine the point when a non-A β biomarker that is potentially earlier than A β accumulation change. However, since the non-A β biomarkers all started with flat trajectories and the first significant change was seen in A β 42 (Fig. 1A), this was likely not a major issue.”

We agree that the sensitivity of the biomarkers potentially plays a role in our findings (e.g. if A β 42 is more sensitive to A β pathology than CSF P-tau to tau pathology). We now address on p. 11, 2nd paragraph:

“In addition to the validity, the sensitivity of the biomarker (for detecting the underlying pathology) may also affect the results. For example, if CSF P-tau is much less sensitive to accumulating tau pathology in the brain than CSF A β 42 to A β , we would find that CSF A β 42 changed earlier even if tau was an earlier pathological mechanism in the brain. We therefore want to note that the identified order of biomarker changes (Fig. 2) refers to the actual biomarkers, which may or may not translate to the order in which the underlying pathology appears in the brain.”

2. The fact that the longitudinal model is entirely based on cross-sectional data should be mentioned also in the methods and results.

Authors' reply:

We have made sure that the word “longitudinal” never is used regarding the present study data. We address our cross-sectional data in the (old and new text):

Abstract:

“Using cross-sectional data from 377 participant in the BioFINDER study...”

Methods, p. 15, 1st paragraph:

“Only cross-sectional CSF, plasma, MRI and PET data were used.”

Results, p. 5, 2nd paragraph:

“Note that all models were fitted using cross-sectional CSF, plasma, and PET data.”

Discussion, p. 12, 2nd paragraph:

“A shortcoming of our study design was that we only included cross-sectional CSF and plasma data.”

“An additional shortcoming was the use of cross-sectional A β PET SUVR as a proxy of time in the disease”

3. Given the novelty of the plasma biomarker results, it would be of interest to add how accurately each of the plasma markers discriminates between amyloid PET-positive and amyloid PET-negative individuals.

Authors' reply:

We agree that this would be interesting, however we think this is out of the scope of the present paper, which already contains a large number of statistical analyses. We also believe that the paper would be less focused if we added analyses on their accuracy to predict amyloid positivity.

4. The authors should mention explicitly on which dataset the mixture modelling was applied that led to the amyloid PET cut-off. According to the Hahn et al paper it is based on the broader BioFinder dataset. It is important to mention how many healthy controls and how many MCI or AD dementia patients this included as a mixture model will give different cut-offs for different cohorts. The authors should also mention whether this cut-off has been validated against a neuropathological standard-of-truth. Although the lack of validation of the amyloid PET cut-off against a gold standard is an issue, a shift in the amyloid cut-off would by no means alter the basic conclusions about the relative longitudinal change between the fluid biomarkers.

Authors' reply:

The Hahn et al paper never determined a cutoff for amyloid positivity using this early A β ROI (continuous data were used for this ROI and amyloid positivity was established using a larger neocortical ROI). In the present paper, we determined the cutoff using mixture modeling in the present population (n=377; 242 cognitively unimpaired, 135 with MCI, none with AD dementia; 151 A β + and 226 A β -; see Table 1). We have not validated this cutoff against neuropathological data.

We now clarify this on p. 18, last paragraph:

“A β status (positive or negative) was established based on mixture modeling statistics (Benaglia, Chauveau et al., 2009) in the present study population using the SUVR data from the early A β accumulating ROI (>0.736 SUVR defined A β positivity). Since it was derived from the present population it has not been validated against neuropathological data. However, this method of establishing A β cut-offs has successfully been used in many previous publications (Bertens, Tijms et al., 2017, Palmqvist et al., 2016, Villeneuve, Rabinovici et al., 2015).”

5. Supplementary figures: For some CSF assays (CSF A β 40, CSF YKL40, CSF NFL) and some plasma biomarker assays (plasma Abeta42, plasma Abeta40, NFL and t-tau), the r2 is very low (0.02-0.08). Even when the p value is significant, for those assays with such a low r2 the spline

model in the main paper may be misleading and it would be better to omit these from the main figures because of the very low amount of variance explained by the model for these assays. Alternatively, it would be fair to highlight in the results that r^2 was very low for these.

Authors' reply:

We now highlight these r^2 values in the Result section in the main manuscript.

Regarding CSF r^2 values (p. 6, 1st paragraph):

“As expected when using A β PET as the dependent variable, the best model fits were seen for CSF A β 42 ($r^2=0.42$) and CSF A β 42/A β 40 ($r^2=0.55$), while poorer fits were seen for CSF P-tau ($r^2=0.30$), T-tau ($r^2=0.25$), neurogranin ($r^2=0.11$), NfL ($r^2=0.11$), YKL-40 ($r^2=0.02$), and A β 40 ($r^2=0.02$).”

Regarding the poorer r^2 values for plasma compared with CSF, this is mentioned on p. 6-7, last-1st paragraph):

“In contrast to CSF, plasma A β 42 and A β 42/40 showed more modest changes over the entire A β accumulation range (about 1 z-score vs about 2 z-scores for CSF; Fig. 3A) and had overall a lesser agreement with A β PET (Plasma r^2 0.07-0.12; CSF r^2 0.42-0.55; Supplementary Fig. 1-2). This lesser agreement was true for all plasma biomarkers compared to the corresponding CSF biomarkers (Supplementary Fig. 1-2), except for plasma P-tau which was more similar to the corresponding CSF biomarker (Fig. 3B).”

and in the Discussion on p. 11, 3rd paragraph, 3rd sentence:

“Overall, the plasma biomarker models had considerably lower r^2 values than the corresponding CSF biomarkers and exhibited smaller dynamic ranges (Supplementary Fig. 1-2 and Fig. 1A-B and 3A)”

Minor comments

1. On p 17 'to a lesser extent' should be expressed in mathematical terms, e.g. whether a weighting was applied.

Authors' reply:

We now specify that it is the average uptake in this ROI (p. 18, 2nd paragraph, 2nd sentence):

“We extracted the average standardized uptake value ratio (SUVR) from brain regions prone to early A β accumulation (Palmqvist et al., 2017), relative the uptake in a previously published composite reference region (Landau, Fero et al., 2015)”

A figure of the ROI is available (open access) in the publication we refer to in the manuscript: Hahn A, Strandberg TO, Stomrud E, Nilsson M, van Westen D, Palmqvist S, Ossenkoppele R, Hansson O (2019) Association Between Earliest Amyloid Uptake and Functional Connectivity in Cognitively Unimpaired Elderly. Cereb Cortex.

We have kept the anatomical description from the original paper but have removed “to a lesser extent” from the following sentence (p. 18, 2nd paragraph):

“This specific region of interest (ROI) was comprised predominantly of the posterior cingulate cortex and precuneus, the subgenual part of the anterior cingulate cortex and smaller parts of the angular gyrus, posterior middle temporal gyrus and middle frontal gyrus (Hahn, Strandberg et al., 2019).”

2a. Figures 1 & 3: Add to the legend that the Z scores are derived from the study cohort itself so that $Z = 0$ corresponds to the mean of the study cohort, rather than the mean of the norm group as would be more commonly the case.

Authors' reply:

We have now added the following sentence to both figure legend 1 and 3:

“To facilitate comparisons between different CSF and plasma biomarkers, the levels have been transformed to z-scores based on the distribution in the present population (i.e., a z-score of 0 corresponds to the mean of the study cohort).”

2b. For correct interpretation of figure 3, the S.D. for the different tests in this cohort would be needed.

Authors' reply:

The mean and SD are provided in Table 1.

3. The value of the early change in A β 40 and A β 42 measured in isolation should be qualified as these measures have a relatively low specificity.

Authors' reply:

We are not completely certain that we understand it correctly, but we interpret it as a comment on the reduced specificity of A β 42 as a marker for amyloid accumulation when it is not measured as a ratio with A β 40, and that we should highlight this lower specificity and the importance of using the A β 42/A β 40 ratio. We now stress this more in the following paragraph, but also refer to studies showing that CSF A β 42 alone can be used to measure early amyloid accumulation (p. 9, last paragraph):

“Even though there are previous studies supporting the use of CSF A β 42 measured in isolation to detect early A β accumulation (Mattsson, Insel et al., 2015, Mattsson, Palmqvist et al., 2019, Palmqvist et al., 2016, Palmqvist et al., 2017), the ratio probably provides a more reliable measure of accumulating A β fibrils and increases its specificity since A β 40 acts as a reference peptide that can for example account for inter-individual differences in CSF concentrations and differences in pre-analytical handling of the samples which otherwise may lead to false positive or negative results using just A β 42 (Janelidze et al., 2016, Lewczuk et al., 2017).”

4a. In the online supplement describing CSF and plasma ptau measurement, the authors should specify for CSF that this pertains to the Eli Lilly assay.

Authors' reply:

The heading for this paragraph has now been changed to (online supplement, heading for 1st paragraph):

“CSF analysis of Lilly P-tau181 and P-tau217”

and the first sentence of this paragraph now reads:

“The Lilly P-tau181 and P-tau217 assays were designed for CSF analysis.”

4b. For CSF, the authors describe antibodies against two phosphorylation sites, it is not clear which of the two is used in the further analyses. For the sake of consistency, on p 16 also mention which phosphorylation sites are targeted for the ptau assays from the other manufacturers. Suppl table 1 lists some of the performance parameters of the two CSF ptau assays but this should also be provided for the plasma ptau assay.

Authors' reply:

We agree that this is not clear and we now clarify that all P-tau assays target the 181 site except for Lilly P-tau217 (Methods, p. 17, last paragraph, 2nd last sentence):

“Note that the Lilly P-tau217 assay was the only one targeting the 217 site, all other P-tau assays targeted 181 (Eli Lilly P-tau, EUROIMMUN P-tau, INNOTEST P-tau, and Lilly P-tau181).”

5. Several tests had a skewed distribution. The authors should mention which statistical test they used to determine the normality of the distribution and whether the distribution was then normal following log-transformation. If the data had a bimodal distribution, then log-transformation would probably not be appropriate.

Authors' reply:

A skewed distribution was determined based on visual inspection of histograms, since many of the statistical methods that test for a (somewhat arbitrary) normal distribution has been criticized. None of the log transformed biomarkers (Plasma NfL, P-tau and T-tau as well as CSF YKL-40 and NfL) had a bimodal distribution (bimodal distributions were only seen for CSF and plasma A β 42 and A β 42/40). After the transformation, normal distributions were seen. We have now added this in the Statistical analysis section (p. 19, 2nd paragraph):

“Plasma NfL, P-tau and T-tau as well as CSF YKL-40 and NfL had skewed distributions (based on visual inspection of histograms) and were therefore natural log-transformed. This resulted in normal distributions for these biomarkers (but one single outlier was observed for plasma NfL).”

2nd Editorial Decision

2 October 2019

Thank you for the submission of your revised manuscript to EMBO Molecular Medicine. We have now received the enclosed report from the referee who was asked to re-assess it. As you will see the reviewer is now supportive and I am pleased to inform you that we will be able to accept your manuscript pending minor editorial amendments.

Please submit your revised manuscript within two weeks. I look forward to seeing a revised form of your manuscript as soon as possible.

***** Reviewer's comments *****

Referee #2 (Remarks for Author):

My comments have been addressed satisfactorily. The only remaining remark relates to the title again: 'increasing' still suggests a longitudinal measure. Maybe '... trajectories in relation to amyloid load ...' would be more appropriate.

2nd Revision - authors' response

9 October 2019

Authors made the requested editorial changes.

Corresponding Author Name: Sebastian Palmqvist and Oskar Hansson

Manuscript Number: EMM-2019-11170-V2